

# Traceable and continuous flow calibration method for gaseous elemental mercury at low ambient concentrations

Teodor D. Andron[1,2], Warren T. Corns[3], Igor Živković[1,2], Saeed W. Ali[1,2], Sreekanth Vijayakumaran Nair[1,2], Milena Horvat[1,2]

[1]Department of Environmental Sciences, Jožef Stefan Institute, Ljubljana, 1000, Slovenia
[2]Jožef Stefan International Postgraduate School, Ljubljana, 1000, Slovenia
[3]PS Analytical, Orpington, BR5 3HP, United Kingdom

*Correspondence to*: Milena Horvat (milena.horvat@ijs.si)

**Abstract.** The monitoring of low gaseous elemental mercury (GEM) concentrations in the atmosphere requires continuous

high-resolution measurements and corresponding calibration capabilities. Currently, continuous calibration for GEM is still an issue at ambient concentrations (1–2 ng m$^{-3}$). This paper presents a continuous flow calibration for GEM, traceable to NIST 3133 Standard Reference Material (SRM). This calibration approach was tested using a direct mercury analyser based on atomic absorption spectrometry with Zeeman background correction (Zeeman AAS). The produced continuous flow of GEM standard was obtained through reduction of Hg$^{2+}$ from liquid SRM NIST 3133 and used for traceable calibration of Zeeman

AAS. Measurements of atmospheric GEM using the calibrated Zeeman AAS were compared with two methods: manual gold amalgamation atomic fluorescence spectrometry (AFS) calibrated with the chemical reduction of NIST 3133 and automated gold amalgamation AFS calibrated using the mercury bell-jar syringe technique. The comparisons showed that factory-calibrated Zeeman AAS underestimates concentrations under 10 ng m$^{-3}$ by up to 35 % relative to the two other methods of determination. However, when a calibration based on SRM NIST 3133 was used to perform a traceable calibration of Zeeman

AAS, the results were more comparable with other methods. The expanded relative combined uncertainty for Zeeman AAS ranged from 8 % for measurements at the 40 ng m$^{-3}$ level to 91.6 % for concentrations under 5 ng m$^{-3}$ using the newly developed calibration system. High uncertainty for measurements performed under 5 ng m$^{-3}$ was due mainly to instrument noise and concentration variation in the samples.

## 1. Introduction

Mercury (Hg) and its compounds are ubiquitous in the environment. Gaseous elemental mercury (GEM) is the main Hg species in the atmosphere (Enrico et al., 2016; Outridge et al., 2018). The relatively high chemical stability of GEM and its long lifetime of about 1 year (Ariya et al., 2015; Si and Ariya, 2018) facilitate its global transport (Koenig et al., 2022). Subsequent oxidation of GEM produces gaseous oxidized mercury (GOM) which can be readily deposited via dry or wet deposition (Phu Nguyen et al., 2019). Deposited GOM can be subsequently methylated to the most toxic Hg species, monomethyl mercury

(Castellini et al., 2022). Assessment and monitoring of chemical transformations and physical processes require high




measurement accuracy for all Hg species to identify transformation pathways and consequently possible effects on human health and biota (Burger and Gochfeld, 2021).

Although GEM is the most abundant Hg species in the atmosphere, its concentrations are still extremely low (1–2 ng m$^{-3}$) (MacSween et al., 2022), thus requiring accurate and precise analytical methods for their determination. Commonly used

analytical methods are based on preconcentration techniques, where atmospheric Hg is sampled on sorbent traps (usually gold, gold-coated silica, or activated carbon traps) (Živković et al., 2020). Following preconcentration, mercury is thermally desorbed from the traps and analysed using atomic fluorescence spectrometry (AFS) (Kamp et al., 2018; Skov et al., 2020) or atomic absorption spectrometry (AAS) (El-Feky et al., 2018). Alongside AFS and AAS, recent analytical methods are also based on laser techniques including laser imaging, detection, and ranging (LIDAR) methods (Fantozzi et al., 2021; Ravindra

Babu et al., 2022) and laser absorption spectroscopy (Srivastava and Hodges, 2022).

A more direct approach for Hg determination in atmospheric samples can be achieved using an atomic absorption spectrophotometer with Zeeman background correction (Zeeman AAS) having a high measurement resolution, such as a Lumex RA-915M. The high measurement resolution is achieved by direct 1-second GEM measurements in the atmosphere. The principles of this method have been described elsewhere (Sholupov and Ganeyev, 1995). This technique is used worldwide

for GEM measurements (Cabassi et al., 2020; Lian et al., 2018) due to its simplicity of operation and absence of a preconcentration step. Zeeman AAS ought to be calibrated at the manufacturer, who suggests a yearly recalibration in the range from low µg m$^{-3}$ to mg m$^{-3}$. However, this might be inappropriate for ambient atmospheric measurements.

In general, the calibration of instruments for GEM determinations can be performed using traceable amounts of Hg$^0$ obtained from i) the reduction of Hg$^{2+}$ certified reference materials (CRMs), ii) a mercury bell-jar calibrator unit, or iii) GEM permeation

tubes (Jampaiah et al., 2019). However, a stable and continuous flow of GEM calibration gas is required for a metrologically proper calibration of instruments for continuous GEM analyses in the atmosphere. Several calibration devices for elemental Hg already exist (albeit for higher concentrations), such as NIST PRIME (Long et al., 2020) and CavKit 10.534/10.536 (Brown et al., 2008) from PS Analytical. They are dynamic continuous GEM calibrators which dilute mercury-saturated air originating from a temperature-controlled reservoir of liquid elemental mercury. Depending on the Hg reservoir temperature, reservoir

flow, dilution flow and pressure, the mercury concentration can be calculated using the Dumarey equation (Dumarey et al., 1985; Ebdon et al., 1989). Mercury saturation in air calculated using the Dumarey equation is an agreed standard used for calibration (ISO 6978-2:2003, ASTM D6850-03) and has been validated using SRMs (Dumarey et al., 2010). In the literature, the lowest concentration of Hg$^0$ validated using a dynamic calibrator is 0.501 µg m$^{-3}$ (NIST Prime), with an expanded uncertainty of 0.018 µg m$^{-3}$ (Long et al., 2020). The CavKit is normally used for concentrations in the µg m$^{-3}$ range (Lopez-

Anton et al., 2010; Wang et al., 2018), but the lowest reported concentration obtained from it is 26.4 ng m$^{-3}$ (Brown et al., 2010) by further diluting the output gas coming from the Hg reservoir. Recently, a primary mercury gas generator (PMGG) was developed for the calibration of GEM measurement systems (Ent et al., 2014). The working principle of the PMGG is based on a dilution of mercury in stripping gas from diffusion cells. The traceability of the PMGG to SI units is established through the determination of diffusion rates, calculated by weighing the diffusion cells at regular time intervals (de Krom et





al., 2021; Ent et al., 2014). Currently, the lowest concentration generated by the PMGG is 0.1 µg m$^{-3}$ with a relative expanded uncertainty of 3 %.

The objective of this paper is to present a simple, cheap, and user-friendly calibration method for the continuous ambient GEM measurements by Zeeman AAS based on the reduction of Hg$^{2+}$ from NIST 3133 SRM. This calibration approach can be used to generate traceable amounts of GEM in the low ng m$^{-3}$ concentration range. An uncertainty budget for the analytical procedure at low concentrations is presented. The performance of Zeeman AAS is compared with manual and automated gold amalgamation systems for GEM determinations.

## 2. Materials and methods

### 2.1. Chemicals and calibration standards

The acids used for the acidification of samples and standard solutions were 30 % HCl (suprapur, Merck, Darmstadt, Germany) and 65 % HNO$_3$ (for analysis, Supelco, Darmstadt, Germany). Type I purified water (electrical resistivity 18.2 MΩ cm; Milli-Q water, Merck, Darmstadt, Germany) was used in all experiments for the dilution of standards. Soda lime (Merck, Darmstadt, Germany) was used for drying the calibration Hg$^0$ gas. The solution of SnCl$_2$ (made from SnCl$_2$2H$_2$O max. 0.000001 % Hg, Merck, Darmstadt, Germany) in HCl was used for quantitative reduction of Hg$^{2+}$ from NIST 3133 SRM standard solutions. NIST SRM 3133 lot No. 160921 (National Institute of Standards and Technology, Gaithersburg, MD, USA) was used for the calibration of Zeeman AAS and manual AFS. The Hg stock solution (1.044 µg g$^{-1}$) was prepared by two consecutive gravimetric dilutions of the initial NIST 3133 SRM (10.004 mg g$^{-1}$) in matching matrix (10 % HNO$_3$ V/V in Milli-Q water). From this stock solution, working standard solutions were prepared daily in 2 % w/w HCl (exact concentrations are provided in the supplement).

For automated AFS, a Hg saturated gas standard was sampled using a gas-tight syringe (Hamilton gas-tight syringe 50 µL, point style 2, 51 mm needle length) from a bell-jar calibrator unit (Tekran® 2505 Mercury Vapor Primary Calibration Unit). The amount of injected mercury was calculated using the Dumarey equation (Dumarey et al., 1985).

### 2.2. Analytical instrumentation

In this paper, we compare the analytical performance of Zeemam AAS with automated and manual gold amalgamation due to their widespread use in monitoring GEM concentrations in the atmosphere. For this purpose, we used a Lumex RA-915M (Lumex Analytics GmbH, D-24558 Wakendorf II, Germany) (Mashyanov et al., 2022) for Zeeman AAS, a Sir Galahad (PS Analytical, Arthur House, Main Rd, Orpington BR5 3HP, United Kingdom) (Rafeen et al., 2020) for automated AFS, and a Brooks Rand Model III (Souza et al., 2020) as the detector for manual AFS. An OLM30B Sampler (Lumex Analytics GmbH, D-24558 Wakendorf II, Germany), equipped with a high-accuracy flow meter, was used for GEM sampling on gold traps.



### 2.2.1. Manual AFS

The manual AFS method is based on a double amalgamation procedure coupled with thermal desorption and AFS detection. Atmospheric GEM samples were collected on a sampling gold trap connected to the OLM30B Sampler. Air was pumped through the sampling gold trap for 30 minutes with a flow of 0.5 L min⁻¹, to match the sampling system of the automated AFS. In the measurement step, the sampling gold trap was immediately transferred to a double amalgamation AFS system. The sampling gold trap was heated for 30 seconds (ramp heating to a maximum of 600°C), and mercury collected on this trap was

released and purged with a flow of argon onto a permanent analytical gold trap kept at room temperature. After heating the analytical gold trap, Hg was thermally desorbed into an argon stream that carried the released Hg vapour into the cell of an AFS detector (Brooks Rand III).

One-point calibration (100 or 250 pg, depending on the concentration in the samples) was used for the calibration of the AFS detector. Gaseous Hg⁰ standards were prepared by the reduction of the appropriate masses of NIST SRM 3133 working

standards in 50 mL of Milli-Q water using 2 mL of 10 % $SnCl_2$ (w/v). The produced Hg⁰ was purged with $N_2$ gas for 4 minutes at a flow of 50 mL min⁻¹, dried through a soda lime trap, and quantitatively trapped on the sampling gold trap. The analytical signal for the calibration standard trapped on the sampling gold trap was obtained in the same manner as described for samples.

### 2.2.2. Automated AFS

The Sir Galahad is an automatic mercury analyser which works similarly to the manual double amalgamation AFS. However,

GEM is preconcentrated on an incorporated single gold trap, thermally desorbed, and analysed in the AFS detector. Air sampling using an external pump was set to 0.5 L min⁻¹ for 30 minutes, resulting in a total volume of 15 L of sampled air (the same as in the case of manual double amalgamation). The preconcentrated GEM is then thermally released in the Ar stream and detected using atomic fluorescence. The automated AFS measures the volume of the sampled air and calculates the results automatically.

The calibration of the automated AFS was performed using GEM saturated gas standard. The gas standard was obtained from a bell-jar gas calibrator unit using a gas-tight syringe. Depending on the bell-jar temperature, the concentration of GEM was calculated using the Dumarey equation (Dumarey et al., 2010). The injected mercury was amalgamated on the gold trap, similarly as in the manual double amalgamation method. The signal for GEM standard was obtained at the AFS detector after thermal release from the gold trap.

### 2.2.3. Zeeman AAS

The Lumex RA-915M is a portable multifunctional atomic absorption spectrometer with Zeeman background correction. Atmospheric GEM is sampled using an in-built pump with a flow of 10 L min⁻¹ (Lumex RA-915M Mercury Analyzer Operation Manual, Mission, Canada, 2015). The Zeeman AAS automatically performs baseline correction at the beginning and end of each analysis by sampling air through the incorporated zero filter. The baseline correction is used to alleviate





possible detector drift. Throughout this work, the baseline correction time was set to 60 seconds. During air analysis, baseline correction was performed every 15 minutes.

The Zeeman AAS instrument is calibrated yearly at the manufacturer's facilities in the range from low µg m$^{-3}$ to mg m$^{-3}$. As this might not be appropriate for low atmospheric GEM measurements, we also performed external calibration as described in Section 2.3. The Zeeman AAS can report GEM values as often as every second. However, a reading resolution of 10 seconds

was used in this work. Lower reading resolutions would greatly increase the uncertainty of integration, while higher resolutions would not significantly decrease the standard deviation. All 10-second signals were integrated using Rapid software for a period of 5 minutes. To eliminate any possible errors due to signal changes, the system blanks were analysed before each external calibration standard, while baseline correction was performed after each standard. Nevertheless, the Zeeman AAS proved to be a very stable detector.

### 2.2.4. External GEM calibration for Zeeman AAS

The principle of external continuous flow GEM calibration was based on continuous reduction of NIST-traceable Hg$^{2+}$ standard solution with SnCl$_2$. The general scheme of the external continuous flow GEM calibration system is shown in Fig. 1. NIST 3133 calibration standard (ranging from 2.5 to 1000 pg g$^{-1}$) and 3 % w/v SnCl$_2$ in 3 % HCl are transferred using peristaltic pumps through a first T-split to a 45 cm Teflon mixing coil to assure quantitative reduction of Hg$^{2+}$ to Hg$^0$. The mixed solution

is then pre-purged in the second T-split with a known flow of Hg-free nitrogen to facilitate gas-phase Hg$^0$ transfer from the liquid solution. The N$_2$ gas carries the solution to an impinger, where the liquid is vigorously purged with the same carrier gas through a quartz frit. The flows of NIST 3133 liquid standards used in this work were between 2.2 and 3 g min$^{-1}$, which was determined by weighing the waste from both the peristaltic pump and the impinger. No noticeable difference was caused by the N$_2$ flow from the impinger. The amount of Hg in the waste exiting the impinger was always below the limit of detection

of the manual double amalgamation system, thus ensuring quantitative transfer to the gaseous phase. In order to obtain the required sampling flow for the external calibration of the Zeeman AAS, the output from the impinger was diluted with a known flow of Hg-free air through a third T-split (Fig. 1). The Hg-free air was assured by the pump of the Zeeman AAS system, being the difference between the flow of N$_2$ coming from the impinger, and the flow of the pump. The Hg filter is a carbon trap similar to the one that the Zeeman AAS device uses for baseline correction. The GEM in the output from the third T-split was

introduced directly to the Zeeman AAS detector and used as an external GEM calibration gas. A soda lime trap placed between the last T-split and the detector removed any moisture coming from the impinger or the air outside the system.

The flow of N$_2$ used was between 3 and 8.3 L min$^{-1}$, depending on the concentration range of the standards used. The flow of N$_2$ was set with the aid of a mass flow controller. For the highest NIST 3133 liquid standard used (1 ng g$^{-1}$), a N$_2$ flow of 8.27 L min$^{-1}$ was necessary in order to assure quantitative release of the Hg$^0$ from the liquid phase. As long as the N$_2$ flow was high

enough to quantitatively purge the Hg$^0$ from the standard, changes in its flow did not bring noticeable changes to the signal.




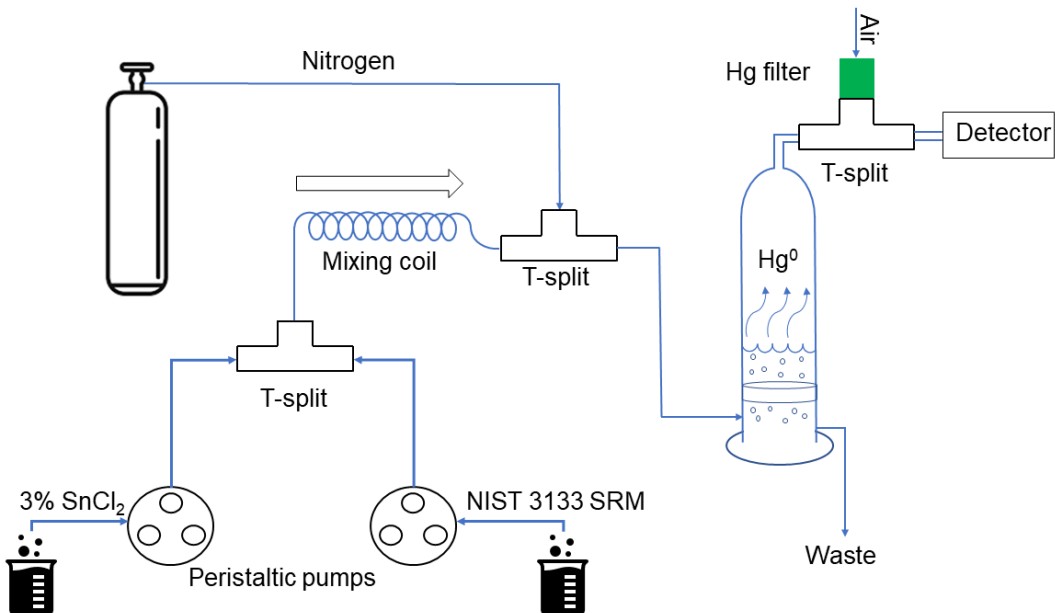

**Figure 1: Scheme of external GEM calibration for Zeeman AAS.**

**2.4. Uncertainty of measurement results for Lumex RA-915M**

The uncertainty of the Zeeman AAS measurement results, calibrated with the external GEM calibration system, was calculated according to the ISO-GUM approach (ISO/IEC 17025, Switzerland, 2017). The uncertainty of each component was calculated following the ISO-GUM rules for the calculation of uncertainty sources. The combined standard uncertainty (at the coverage factor of 1) was calculated following the general rules for the propagation of uncertainty for a mathematical model expressed

in the form of additions and/or multiplications. The identification of uncertainty components was based on the mathematical model (Eq. (1) and (2)) used for the calculation of GEM concentrations from the detector's analytical signal.

$$ C_S = \frac{A'_S - i}{m} \tag{1} $$

$$ C_{std\ gas} = C_{NIST\ 3133} \times \frac{Q_{aq}}{Q_{inflow}} \tag{2} $$

where:

$C_S$ is the concentration of the air sample [ng m$^{-3}$]

$m$ is the best-fit gradient of the calibration curve (slope)

$i$ is the intercept of the calibration curve





$A'_S$ is the signal of the sample after blank subtraction [ng m$^{-3}$]

$C_{std\ gas}$ is the concentration of the produced gas standard, and was used to calculate $m$ and $i$ [ng m$^{-3}$]

$C_{NIST\ 3133}$ is the concentration of the liquid standard [ng g$^{-1}$]

$Q_{aq}$ is the flow of liquid SRM which is introduced by the peristaltic pump in the T-split before the impinger [g min$^{-1}$]

$Q_{inflow}$ is the flow of air and nitrogen that enters the detector [m$^3$ min$^{-1}$].

Based on the model, a fish-bone diagram (Ishikawa diagram) was constructed and is presented in Fig. 2. Proper construction of the fish-bone diagram was an important step for the uncertainty evaluation, as it showed the influences of different components on the combined uncertainty. Fish-bone diagrams for the manual and automated AFS systems are presented in the supplementary information (Figures S.1 and S.2).

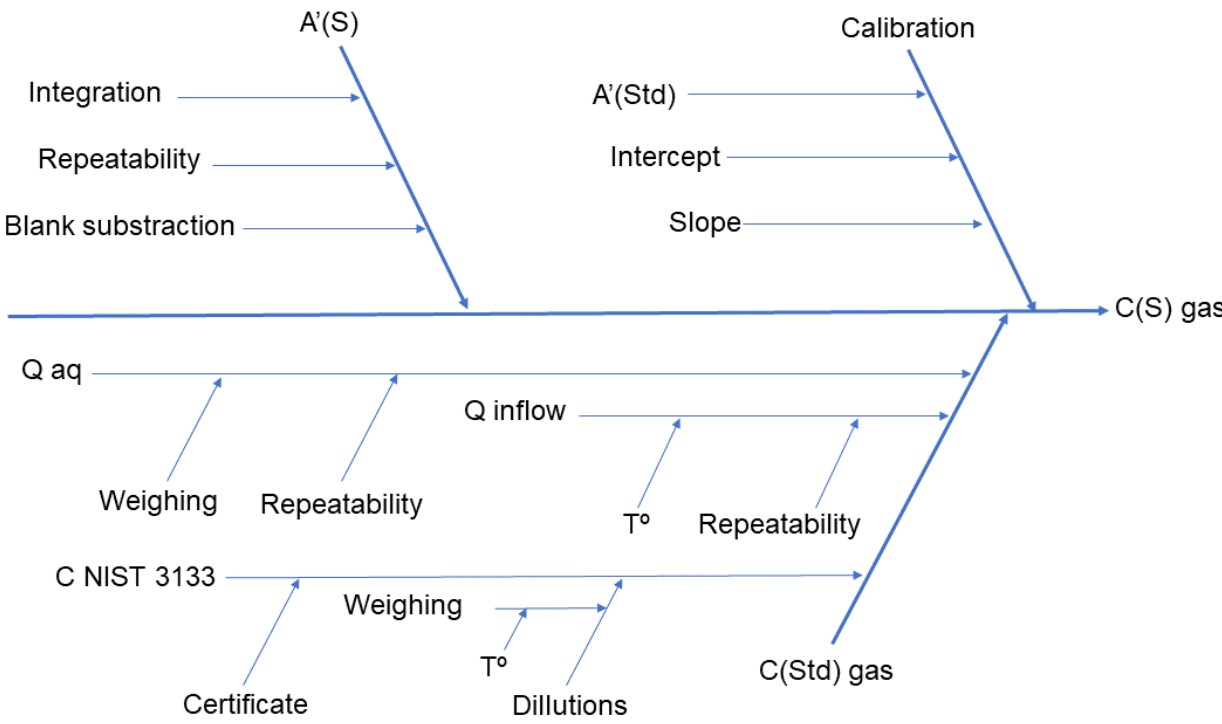


**Figure 2: Fish-bone diagram for the uncertainty of measurement results in the case of sample analysis with the Zeeman AAS.**

The uncertainty of the calibration curve was estimated following EURACHEM guidelines (EURACHEM/CITAC Guide,

Quantifying Uncertainty in Analytical Measurement, United Kingdom, 2012):





$$u_x = \frac{S_y}{m} \times \sqrt{\frac{1}{k} + \frac{1}{n} + \frac{(y - \bar{y})^2}{m^2 \sum(x_i - \bar{x})^2}} \qquad (3)$$

where:

$u_x$ is the standard uncertainty brought by calibration for the value $x$

$S_y$ is the standard deviation on the y axis

$m$ is the calculated best-fit gradient of the calibration curve

$k$ is the number of replicated measurements on the sample

$n$ is the number of paired calibration points

$y$ is the mean value of the replicates of the sample

$\bar{y}$ is the mean of the $y$ values for the calibration standards

$x_i$ is the position of the standards on the x axis

$\bar{x}$ is the mean of the $x_i$ values.

All uncertainty components were calculated as type A uncertainty, except for the weighing and temperature, which were calculated as type B uncertainties. The expanded combined relative standard uncertainty was calculated by multiplying the relative combined standard uncertainty by a coverage factor of 2:

$$U_{ex,r} = u_{r,c} \times k \qquad (4)$$

The contribution index ($\%u_i$) of individual uncertainty components was calculated as follows:

$$\%u_i = \frac{u_i^2}{\sum_i^n u_i^2} \qquad (5)$$

where $n$ is the number of uncertainty components used to calculate the relative combined standard uncertainty.

**2.5. Data analysis and statistical evaluation**

Uncertainty evaluation was performed following ISO-GUM guidelines (ISO/IEC Guide 98-3:2008) and was calculated using Microsoft Excel 2019. Appropriate statistical tests (analysis of variance (ANOVA), repeated measures ANOVA, regression analysis) were performed using SigmaPlot 14.0. The statistical significance was set to α value of 0.05.


## 3. Results

### 3.1. External calibration for the continuous mercury analyser

The stability of the external calibration of the Zeeman AAS signal is shown in figure S.3. Although in all following figures the values on the y axis are expressed as Hg concentrations and not as raw analytical signals, we use the expression 'calibration curve' in these figures to emphasize the necessity of additional external calibration of already-calculated Hg concentrations. Therefore, the values on the y axis can be considered equivalent to the analytical signal. For an external calibration presented in figure S.3 was performed using corresponding liquid standards with concentrations between 5 and 55 pg g$^{-1}$ as described in Section 2.2.4. The system was calibrated from the highest to the lowest concentration to test whether Hg leftover remained in the system after each calibration standard. This was confirmed by the absence of the systematic differences between the blanks after each standard.

The repeatability of both blanks and standards are comparable. The standard deviation varied between 0.32 and 0.43 ng m$^{-3}$ for a reading resolution of 5 seconds, which indicates that there was a stable outflow from the calibration setup (Fig. 1). The background correction was done after each standard in order to avoid any uncertainty arising from the drift of the signal. For each blank and standard, an integration time of 5 to 6 minutes was used. As all the results reported throughout this manuscript were calculated based on multiple-point calibration curves, the blank measured before each standard was used for subtraction. The concentration output was also highly dependent on the autozero done by the Zeeman AAS system. In order to eliminate any drifts that may occur during calibration, autozero was applied after each pair of blank-standard detection.

Calibration curves were also performed when the standards were introduced in random order of concentrations to demonstrate the absence of bias due to hysteresis. The slope and regression coefficients of the calibration curves did not differ. An example of this randomized calibration curve was used to calculate the results in Section 3.3.3.

### 3.2. Comparison of factory calibration against a known concentration generated by NIST 3133 at different concentration levels

The linearity of the detector's response was checked using a wide range of concentrations, ranging from 1.76 to 176.8 ng m$^{-3}$, as shown in Fig. 3A. The corresponding concentrations of Hg in the NIST 3133 liquid standards were between 10.4 and 1044 pg g$^{-1}$. All standard solutions were analysed daily under the same experimental conditions for external GEM calibration of the Zeeman AAS (Section 2.2.4). The overall slope of the calibration curve was close to 1 (Fig. 3). However, in the lower calibration range, the slope was considerably lower (0.681; Fig. 4B), thus confirming the requirement for additional external calibration at low concentration levels (<15 ng m$^{-3}$). The five of the lowest calibration standards (Fig. 3B) fell within the 95 % confidence interval of the overall calibration curve calculated based on all 10 calibration standards. If only one calibration standard were measured in this low concentration range, samples measured in this region would be underestimated.

This difference in response at lower concentrations was reproducible and constant throughout this work. Even though it is not known why the difference occurred at low concentrations, the lower response was observed during sample analysis of GEM.





This indicates that there was not an issue with the external calibration but rather a bias from the Zeeman AAS. Factors such as spectral emission profiles, concentrations of analyte, and magnetic field strength can influence the performance of Zeeman background correction (Ganeyev and Sholupov, 1992).


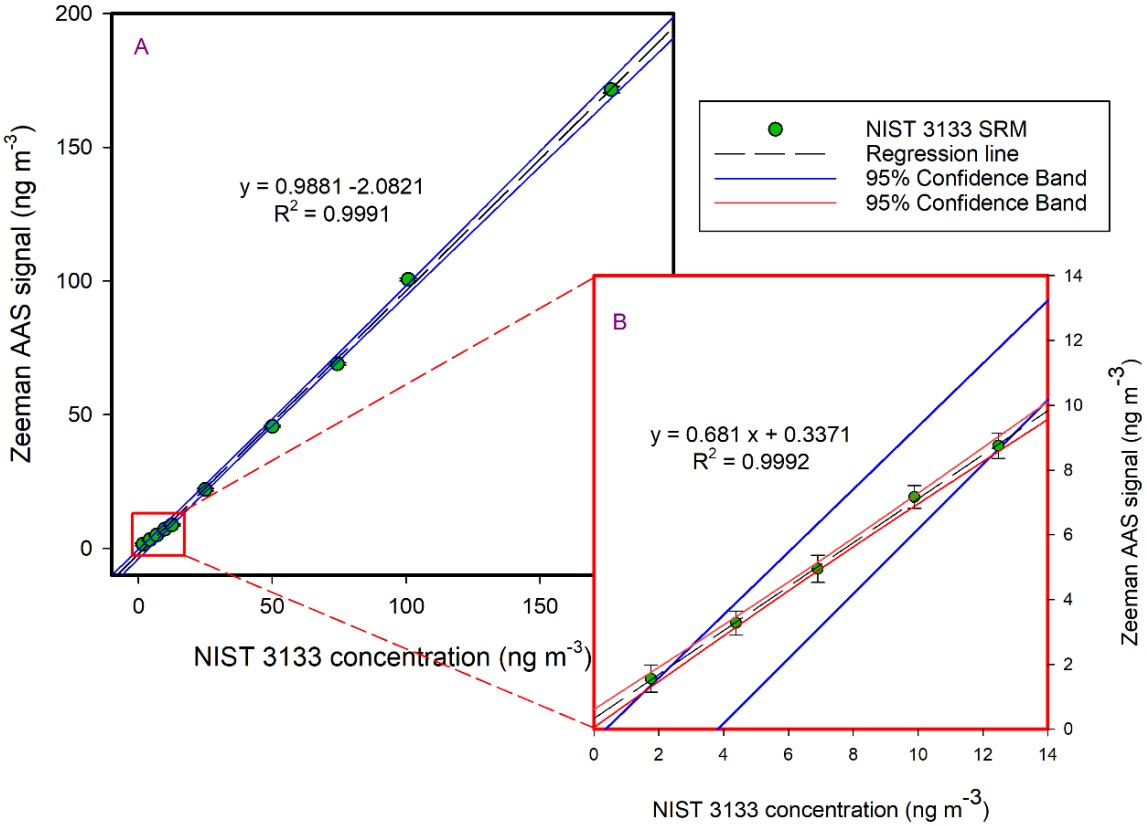

**Figure 3: (a) Analysis of NIST 3133 standards in the range of 1.76–176.8 ng m$^{-3}$; (b) Analysis of NIST 3133 standards in the range of 1.76–12.5 ng m$^{-3}$. The x axis is the theoretical concentration of the NIST 3133 SRM, and the y axis is the calculated concentration by the Zeeman AAS device, proportional to the signal.**


Furthermore, the external calibration curve (Fig. 3) was plotted along with the internal calibration curve determined by the manufacturer (Fig. 4) during yearly service. The calibration standards used by the external calibration system were well within the confidence interval of the manufacturer calibration, and the corresponding slopes were parallel. Therefore, the application of the internal calibration curve for the determination of high GEM concentrations was valid and gave accurate results. For the

lower concentration region in Fig. 3B, proper external calibration must be performed.



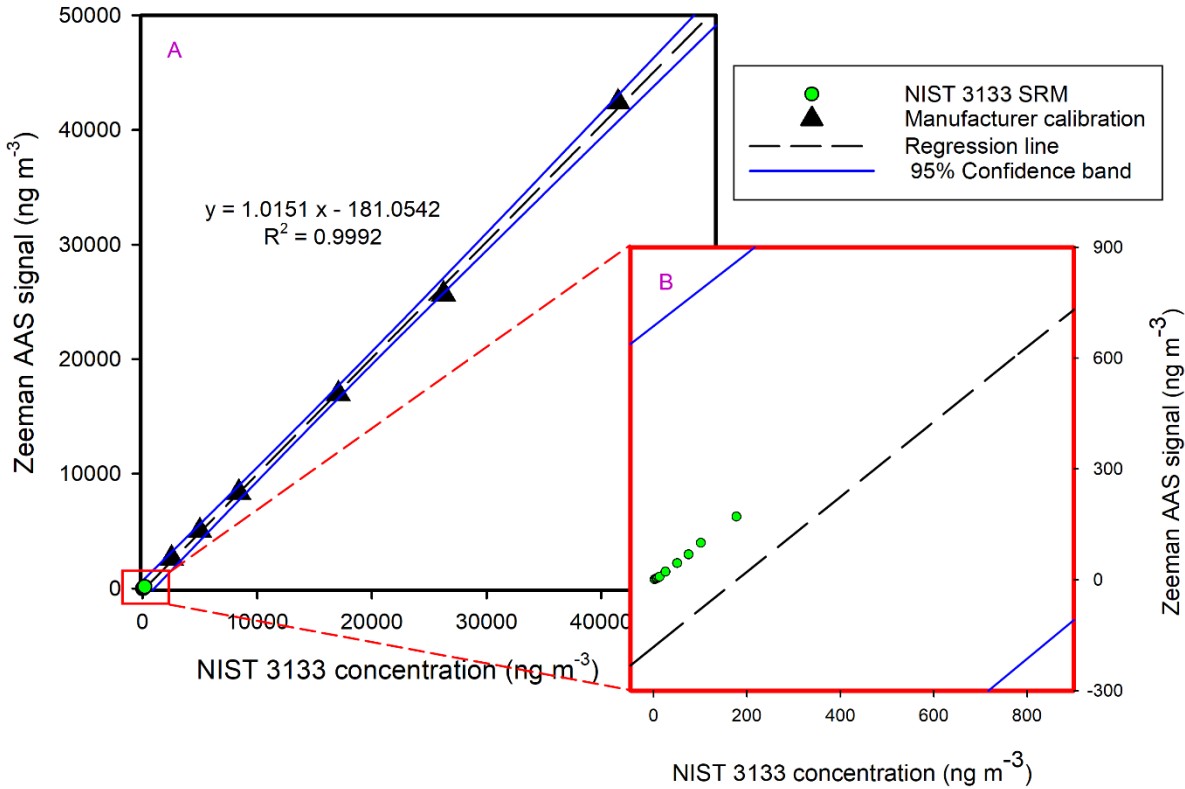

**Figure 4: (a) Manufacturer calibration in the range of 2532–41476 ng m$^{-3}$; (b) NIST 3133 standards in the range of 1.76–176.8 ng m$^{-3}$.**

### 3.3. Comparison between AAS, automated double amalgamation AFS, and manual double amalgamation AFS

In order to validate the external calibration method, Zeeman AAS was compared with automated AFS and manual AFS at different calibration ranges. The measurements were performed at high, medium, and low concentrations, defined by the following criteria. High concentration (around 40 ng m$^{-3}$) measurements were those measurements where no difference was observed between the three, with comparable results and combined extended uncertainties. In medium-concentration ranges (5–10 ng m$^{-3}$), there were differences between the results obtained with the manufacturer-calibrated Zeeman AAS and the other two methods, with the combined extended uncertainties being low enough to back up this statement. Low concentration ranges (<5 ng m$^{-3}$) were concentrations where differences between the methods were observed, but the combined extended uncertainties were too high to actually state that there was a difference.

### 3.3.1. High concentration measurements

Measurements at approximately 40 ng m$^{-3}$ were performed using the above-mentioned methods. No noticeable differences between the three methods were observed, with the combined extended uncertainties varying between 2.2 and 9 %. Even after





recalculating the results based on the external calibration, the results from all three methods were comparable. The average difference between the raw results from the Zeeman AAS system and the ones recalculated using the external calibration method for this set of data was 3.85 %. The main contributor to the combined uncertainty in the case of Zeeman AAS was the
repeatability of the sample, bringing on average a contribution of 76.4 %.

Through evaluating the uncertainty ranges at a coverage factor of 2, it cannot be stated that there was a difference between the three methods. The same statistical evaluation performed in the previous section for low concentrations, and the one-way repeated measures ANOVA with post hoc Bonferroni test for pairwise multiple comparison, also showed no difference between the three methods. The same test was performed with the data calculated with the external GEM calibrator system,
and there was no statistical difference.

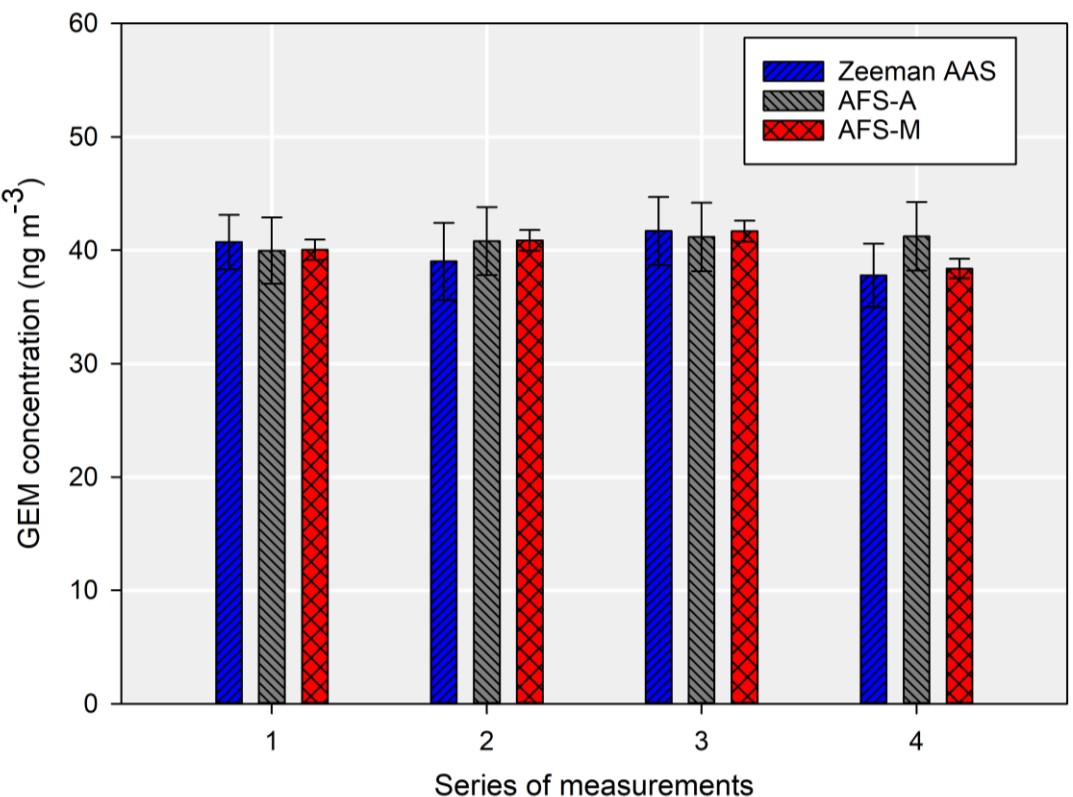

**Figure 5: Comparison between the Zeeman AAS, AFS-A, and AFS-M. The error bars are the combined uncertainties for k = 2.**

### 3.3.2. Medium-concentration measurements

Measurements under 10 ng m$^{-3}$ showed that the Zeeman AAS was underestimating the results even though the manual AFS
and automated AFS systems were comparable, with relatively low uncertainties (Fig. 6). The long-term measurements showed that the difference between the Zeeman AAS system and two AFS methods was, on average, 35.1 %.





Being in the low concentration range for the Zeeman AAS, the relative standard deviation varied from 5.6 % at the highest concentration (8.5 ng m$^{-3}$) to 9 % at the lowest one (4.3 ng m$^{-3}$). In all cases, the contribution from sample repeatability was the highest contributor to the combined uncertainty, which can be seen in Fig. 5. The uncertainty contribution coming from

the gaseous standard was very low (<4 % for this series of measurements).

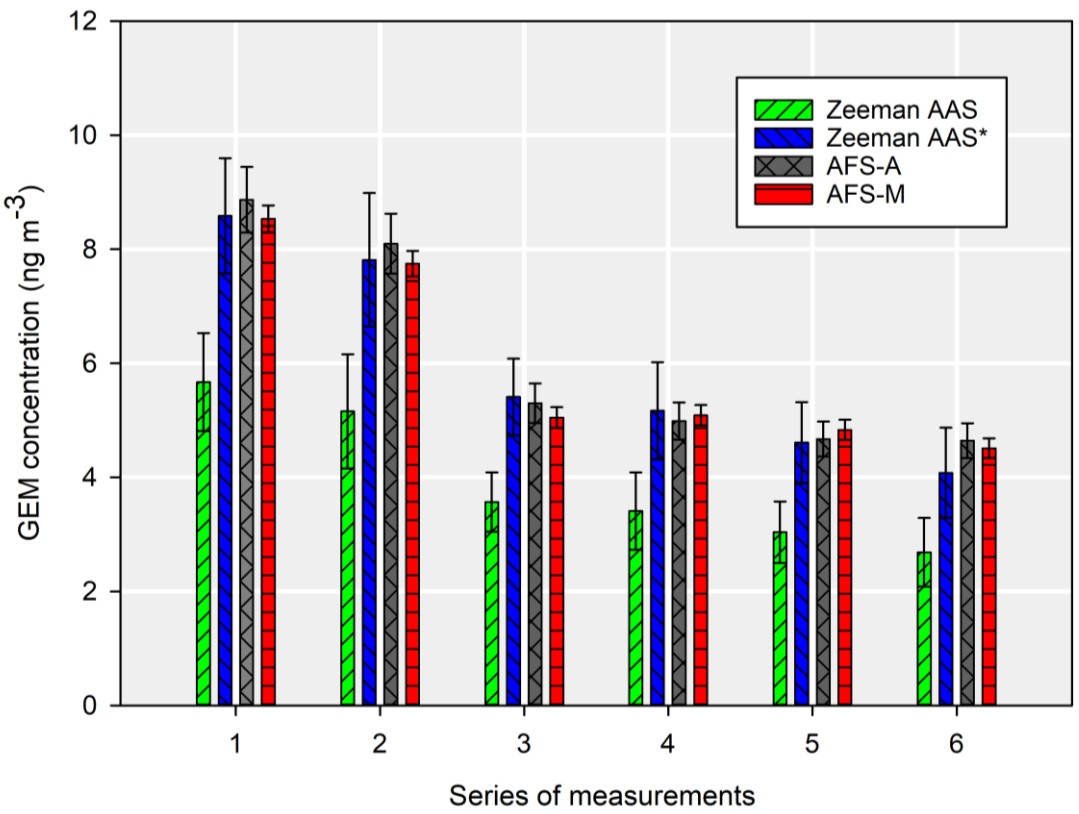

**Figure 6: Comparison between the Zeeman AAS, AFS-A, and AFS-M. The error bars are the combined uncertainties for k = 2. The * stands for externally calibrated Zeeman AAS. For the direct mercury analysers calibrated at the manufacturer, the error bars represent twice the standard deviation, while in the other cases they are represented by the combined expanded uncertainty for a**

**coverage factor of 2.**

The uncertainty intervals greatly overlapped, which indicates that there was no difference between the manual and automated AFS methods. It was the same for the AAS system calibrated with our setup, but not for the raw data obtained with the manufacturer calibration. In order to further confirm whether the differences existed, one-way repeated measures ANOVA

was performed, with post hoc Bonferroni test for pairwise multiple comparison. The results indicated that there was no difference between automated and manual AFS (p = 1), but there was a difference between the results from the first two methods and the results from the Zeeman AAS calibrated at the manufacturer (p < 0.001). Between the first two methods and the results from the Zeeman AAS system calibrated with the dynamic setup, there was no difference (p = 1).





### 3.3.3. Low concentration measurements

Another comparison was made at a lower concentration using two Zeeman AAS systems of the same series and the automated AFS system (Fig. 7). The AAS devices showed similar responses. The uncertainty due to sample repeatability was much higher due to the lower concentrations, and it increased the combined uncertainty by up to 98.6 % for a coverage factor of 2. As seen in Fig. 8, the repeatability of the sample was still the main contributor to the extended combined uncertainty (contribution up to 95 %), like in previous cases.

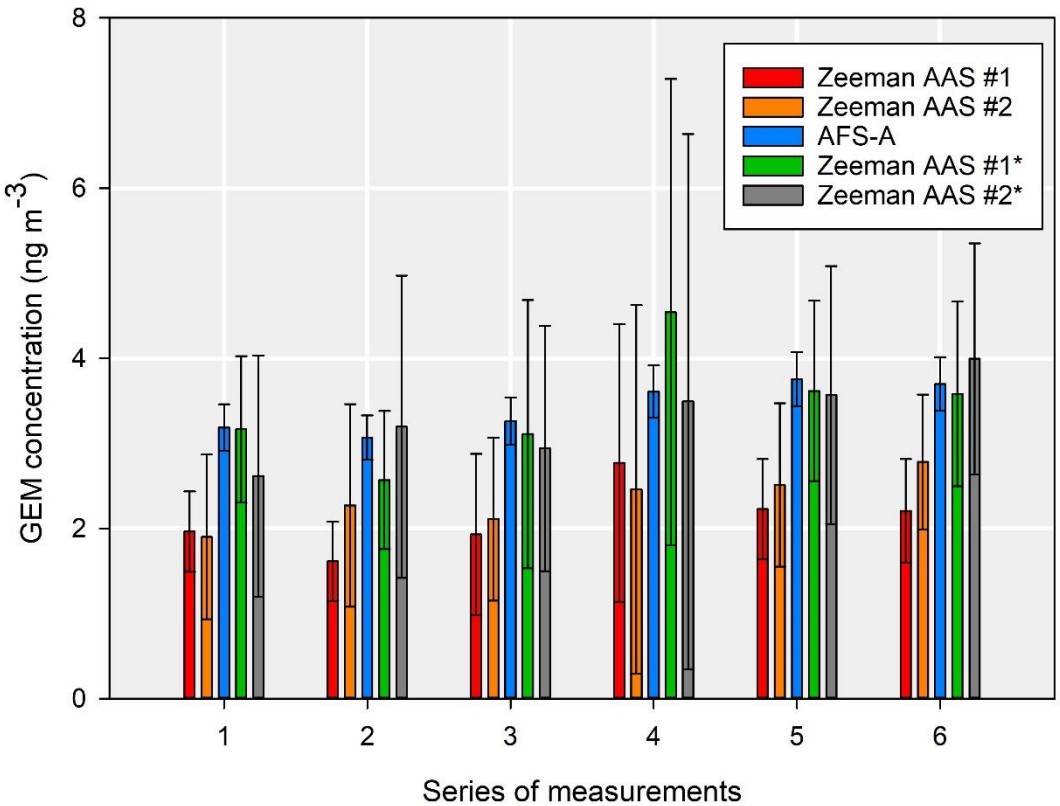


**Figure 7: Comparisons between two direct mercury analysers and the atmospheric mercury analyser. The \* stands for externally calibrated Zeeman AAS. For the direct mercury analysers calibrated at the manufacturer, the error bars represent twice the standard deviation, while in the other cases they are represented by the combined expanded uncertainty for a coverage factor of 2.**

It is important to note that the high uncertainty contribution brought by the repeatability of the sample in the case of Zeeman AAS was not entirely due to its performance but also due to the variability of the concentration of the sample, which in this case was atmospheric air. Even slight changes at these low concentrations result in very high standard deviations. The standard



deviations of calibration standards and blanks in this concentration range were always under 0.5 ng m$^{-3}$, while for samples it

was as high as 1.1 ng m$^{-3}$. This high contribution from the repeatability of the sample is to be expected, as the limit of

quantification is 3.2 ng m$^{-3}$.

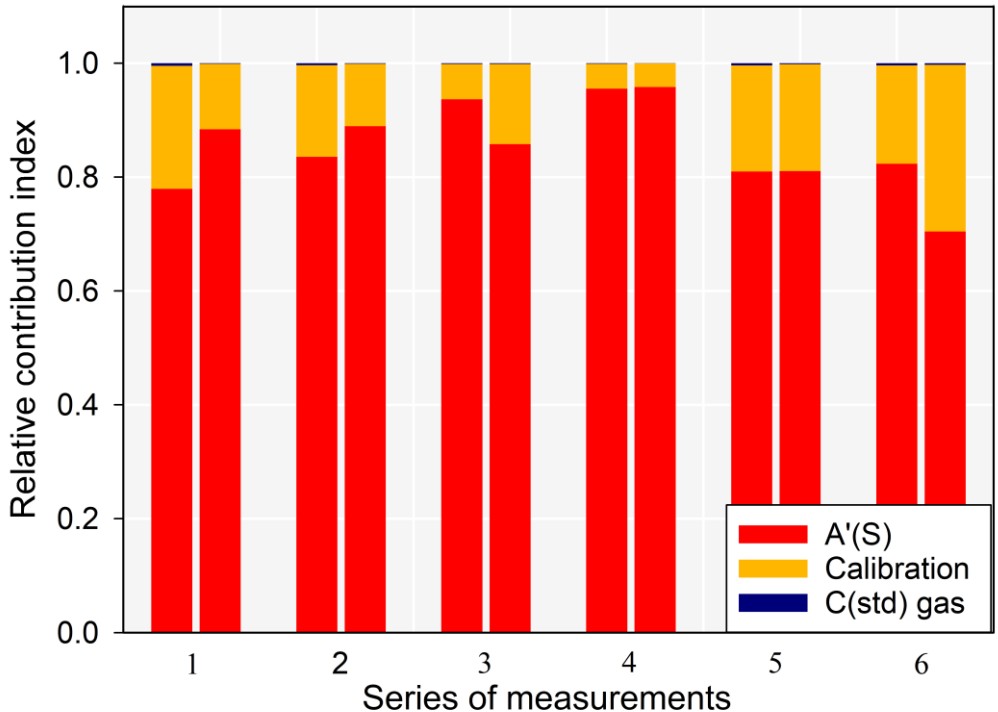

**Figure 8: Relative contribution indexes to the uncertainties for the Zeeman AAS systems externally calibrated.**

Due to these high uncertainties of measurement results, it cannot be stated that there was a difference between the results

obtained from the manufacturer-calibrated Zeeman AAS systems and the automated AFS system, or between the results

obtained from the manufacturer-calibrated Zeeman AAS systems and the externally calibrated Zeeman AAS. This is due to

the fact that the uncertainty ranges overlapped. Even so, the previously used one-way repeated measure ANOVA shows that

there was a difference between the results obtained with the manufacturer-calibrated Zeeman AAS and the automated AFS

($P < 0.001$) and that there was a difference between the results obtained with the manufacturer-calibrated Zeeman AAS and the

externally calibrated Zeeman AAS systems ($p < 0.001$). There was no difference between the automated AFS and the

externally calibrated Zeeman AAS systems ($p = 1$).

Additionally, one-way ANOVA with Holm-Sidak pairwise comparison was performed, and, in most of the cases, there was

no statistical difference between the externally calibrated Zeeman AAS systems and the automated AFS, as can be seen in the

supplementary material (Table S.1). We tested statistical differences between corresponding individual values that were


averaged to calculate the results shown in Fig. 8. The results of one-way ANOVA showed a significant difference between the manufacturer-calibrated and externally calibrated Zeeman AAS ($p < 0.001$ in all cases, except between Zeeman AAS#2 and Zeeman AAS#2* from the 4th series of measurements, $p = 0.004$). There was no significant difference between the two externally calibrated Zeeman AAS systems ($p > 0.05$). The AFS-A was not taken into account for this statistical test, as each result is based on a single data point.

**4. Discussion**

It is difficult to assess why the Zeeman AAS system underestimated the results for low concentrations, as shown in Section 3.3.1. Most likely it is associated to the extrapolation of the regression function obtained from the calibration curve performed at the manufacturer. In the calibration certificate received for the device used in this work, it is stated that the calibration was performed in the range of 2.63–42.4 $\mu$g m$^{-3}$. An extrapolation from these high concentrations to the ones presented in Section

3 can lead to substantial systematic errors. Even so, the Zeeman AAS system performed very well for concentrations of around 40 ng m$^{-3}$.

For the comparison experiments, the repeatability of the samples could not be taken into account for the two preconcentration methods. This is due to the fact that sample homogeneity was not assured from one sampling period to the next. For this reason, additional statistical tests were performed as stated in Sections 3.3.1 and 3.3.2.

The preconcentration methods used for comparisons are sampling and analysing all mercury species in the atmosphere. Along with GEM, GOM and particle-bound mercury (PBM) are also present in the atmosphere (Schroeder and Munthe, 1998). During the thermal desorption of mercury from gold traps, GOM and PBM are reduced, in the end being analysed as GEM as well. Reactive mercury (RM) is the sum of GOM and PBM, and its concentration is usually under 100 pg m$^{-3}$ in continental areas (Pierce et al., 2018; Zhou et al., 2018). Although speciation analysis has not been done, concentrations higher than 100 pg m$^{-3}$

RM were not observed at the sampling site when using a 2537B/1130/1135 Tekran speciation system (Gustin et al., 2013). These concentrations would not affect the results of the comparisons done in this work, as they are almost 3 orders of magnitude lower than was analysed.

Future work should be focused on improving the calibration setup in order to obtain higher concentrations of GEM. The limiting factor in this work was the high flow needed for quantitative removal of the dissolved gaseous mercury from the

solution introduced in the impinger. The highest purging flow used was 8.27 L min$^{-1}$, which managed to quantitatively purge the mercury from a 1 ng g$^{-1}$ Hg$^{2+}$ solution introduced at a rate of 2.26 g min$^{-1}$ into the impinger. Higher flows were not tested due to safety reasons, but, at the moment, quartz is the best material for handling mercury as it does not retain it, compared to other materials which are more durable, such as stainless steel.



## 5. Conclusions

Continuous flow calibration for GEM was established through a simple, cheap, and easily assembled setup. All measurements are traceable to NIST 3133 CRM, and other liquid CRMs of Hg$^{II}$ can be used. The setup ensured stable and repeatable signals even at very low concentrations suitable for the atmospheric continuous measurements of GEM. The main drawback of the calibration setup is that it cannot be used for very polluted measurements, as higher flows of purging gas would be required, which implies considerable pressure inside the system. The calibration method was successfully tested using a commercially

available Zeeman AAS system, and the results were compared with an automated gold amalgamation AFS system calibrated with the Dumarey bell-jar gas standard and with manual AFS calibrated with NIST 3133 SRM. Although the Zeeman AAS system calibrated at the manufacturer underestimated concentrations under 10 ng m$^{-3}$, the newly developed calibration system corrected this. The two AFS methods were in good agreement, with relatively low uncertainties (<9 % for a coverage factor of 2). Improvements to the continuous flow calibration setup may result in obtaining a broader range of concentrations for the

gas standard obtained from NIST 3133 SRM.

**Supplement.** The supplement related to this article is available online at:

**Data availability.** The dataset used in this paper is available upon reasonable request.


**Author contributions.** TDA was responsible for conceptualization, methodology, formal analysis, investigation, data curation, writing – original draft, visualization. WTC was responsible for conceptualization, validation, resources, writing – review & editing, supervision. IŽ was responsible for data curation, writing – review & editing, visualization. SWA was responsible for formal analysis, writing – review & editing. SVN was responsible for formal analysis, writing – review &

editing, visualization. MH was responsible for conceptualization, writing – review & editing, supervision, project administration, funding acquisition. All listed authors are responsible for the review and editing process.

**Competing interests.** The contact author has declared that none of the authors has any competing interests.

**Acknowledgements.** The authors would like to thank Iris de Krom and Adriaan van der Veen for assistance with statistical analysis, Matthew A. Dexter for valuable insight on the manuscript, and Jože Kotnik for technical assistance.

**Financial support.** The authors acknowledge the financial support from the European Commission (GMOS-Train project grant agreement ID: 860497) through the Marie Skłodowska-Curie Initial Training Network and the Slovenian Research

Agency (ARRS) through program P1-0143. Project 19NRM03 SI-Hg has received funding from the EMPIR programme co-financed by the Participating States and from the European Union's Horizon 2020 research and innovation programme.



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
