# Peer review of "Traceable and continuous flow calibration method for gaseous elemental mercury at low ambient concentrations"

_Atmospheric Measurement Techniques, 2023_

## Referee Comment (RC3)

[referee-annotated manuscript omitted]

---

## Author Response (AR1)

Dear Editor,

We have addressed all comments by the two anonymous referees. The responses are presented below, together with corresponding changes in the manuscript. In addition, we have addressed the technical comment raised by the anonymous referee #2 in his Short report. Aside from the changes suggested by the anonymous referees, we have also corrected some typos and added a new acknowledgement, all tracked.

**Reviewer 2**

Comment 1: The sentence "The Hg-free air was assured by the pump of the Zeeman AAS system, being the difference between the flow of N2 coming from the impinger, and the flow of the pump." (line 148) is unclear for me. It is obvious, that the Hg free air was produced by the pump of the Zeeman AAS system. What difference and what pump is meant in the second part of the sentence? Would it be posible to re-formulate this sentence?

Author's response: Indeed, the sentence will be changed.

Changed in the manuscript: The sentence has been replaced with "As the $N_2$ flow is lower than the flow of the pump inside the Zeeman AAS, a T-piece was placed between the impinger and the device. This allows for the air from outside to enter the system in order to compensate for difference in flows and pressure. The air passes through a Hg filter to assure that it does not interfere with the analysis."

Comment 2: What is the precision of setting/stability of the air flow arising from the entrance pump of Lumex?

Author's response: The flow was measured during each experiment and the variation of the flow is <1% which was taken into account when calculating the combined uncertainty of measurement results.

Changes in the manuscript: none

Comment 3: The precise reading of the Lumex unit is pressure dependent. Let's consider an atmosphere with a constant air Hg concentration. At a higher pressure, a higher absolute amount of Hg enters the Lumex measuring cell and the instrument records a higher absorbance.

Was this problem considered in any way in the calibration method discussed? Has the pressure of the gaseous mixture entering the Lumex been monitored or measured in any way?

Author's response: Indeed, the pressure plays a vital role in the actual amount of the analyte entering the system. The pressure was checked in order to be sure that it does not change compared to when it is sampling atmospheric air. This is also one of the reasons why we placed another T-split before the device, in order to compensate for the differences between the flows and pressure that might me between the flow/pressure of the pump and the outflow coming from the impinger.

Changes in the manuscript: none

**Reviewer 1**

Comment 1 (line 78): Dot is missing.

Change in the manuscript: From "SnCl$_2$2H$_2$O" to "SnCl$_2$•2H$_2$O".

Comment 2 (line 78): Scientific number annotation will improve readability.

Change in manuscript: From "max. 0.000001 % Hg" to "max. $10^{-6}$ % Hg"..

Comment 3 (line 89): This is the preconcentration technique and not the analytical method to compare. Isuggest you add AFS.

Change in manuscript: From "In this paper, we compare the analytical performance of Zeeman AAS with automated and manual gold amalgamation due to their widespread use in monitoring GEM concentrations in the atmosphere." to "In this paper, we compare the analytical performance of Zeeman AAS with automated and manual gold amalgamation, followed by AFS quantification due to their widespread use in monitoring GEM concentrations in the atmosphere."

Comment 4 (line 95): Perhaps replace detection with measurement or quantification.

Changes in the manuscript: The word detection was replaced with "quantification".

Comment 5 (line 103): Calibration uncertainty cannot be calculated with 1 point calibration.

Author's response: We calculated the uncertainty of the single point calibration for the Manual AFS as the standard uncertainty which was calculated from the repeatability of the standard. The system has been checked previously for its linearity in this concentration range. As the single point calibration was performed at ambient concentration which was close to the concentration of the ambient sample, the standard uncertainty due to calibration was calculated as the RSD of the calibration standard.

Changes in manuscript: Added citation (Ma et al., 2012).

Comment 6 (Line 117): How many calibration points?

Author's response: 8 calibration points.

Change in the manuscript: From "The calibration of the automated AFS was performed GEM saturated gas standard". to "The calibration of the automated AFS was performed using 8 calibration points of GEM saturated gas standard."

Comment 7 (line 130): replace with sampling frequency

Author's response: The sampling frequency has not been changed throughout the experiments. At first glance it might indeed seem to be the sampling frequency but it is just the reading resolution or acquisition rate.

Changes in manuscript: none

Comment 8 (line 134): This contradicts to the observation of impaired signals with longer sampling. Unless there's sensitivity drift the longer the sampling the lower should be the RSD. Can the authors clarify, please. Is the ambient Hg changing while sampling?

Author's response: This has been done due to the fact that there is a drift when the autozero is done at higher intervals of time. In order to not have this issue we did bracketing sequences of autozero – blank – standard and so on. The Hg concentration changes when sampling ambient air.

Changes in the manuscript: none

Comments 9-10 (147-148): Not clear

Author's response: This part will be changed in accordance also with Reviewer 2 comments.

Changes in the manuscript: The sentence has been replaced with "As the $N_2$ flow is lower than the flow of the pump inside the Zeeman AAS, a T-piece was placed between the impinger and the device. This allows for the air from outside to enter the system in order to compensate for difference in flows and pressure. The air passes through a Hg filter to assure that it does not interfere with the analysis."

Comment 11 (Figure 1) : "T-piece" it is joining flows so better avoid using "split".

Changes in the manuscript: "T-split" has been replaced with "T-piece" in figure 1, as well as in the rest of the manuscript.

Comment 12 (line 213). What is data was analysed with ANOVA?

Author's response: The tests were used to determine whether significant differences exist between atmospheric Hg concentration obtained by different analysers.

Changes in the manuscript: Aded the specification "The tests are mentioned throughout the manuscript along with the data on which the tests were applied."

Comment 13 (line 219): Why not using the term calibration plot?

Changes in the manuscript: "Calibration curve" has been replaced with "Calibration plot".

Comment 14 (line 224). Saturation issues will be missed with this approach. The calibration standards need to be randomised.

Author's response: We agree, in the line 234 we state that this was tested.

Changes in the manuscript: none

Comment 15 (line 225). Can this information be supplied please. Usually blanks would have lower sd but higher RSD.

Author's response: Blank data, together with corresponding SD and RSD%, will be supplied with the comments in excel format. Indeed RSD% is higher for blanks than for standard especially when comparing to higher concentration standard. The statement was intended for the standards in the lower concentration range as it can be seen in the attached data. (blank data was provided in the interactive discussion).

Changes in the manuscript: It has been stated that this is the case only in the case of low concentration standards.

Comment 16 (line 227). This contradicts to the statement about the stability of the analyser. Blank drift modeling would be more efficient than single point corrections.

Author's response: Single point calibration was used only in the AFS-M method as its linearity has been continuously verified before and during these experiments in this concentration range. For the rest it was multiple point calibration.

Changes in the manuscript: none

Comment 17 (line 233): "Re-calibration"?

Changes in the manuscript: From "Calibration curves were also performed when the standards were introduced in random order of concentrations to demonstrate the absence of bias due to hysteresis." to "Recalibration was also performed when the standards were introduced in random order of concentrations to demonstrate the absence of bias due to hysteresis."

Comment 18 (line 234). was statistical test for difference performed? If yes, the results should be stated here.

Changes in the manuscript: The next section was added: "Parallel Lines Analysis (PLA) was performed between two calibration curves which were in the same concentration range (one with randomised calibration points and one calibration points analysed from the lowest to the highest concentration) and there is no statistical difference between them with $P = 0.1995$ for the slopes and $P = 0.5493$ for the intercepts. The equation for the randomised calibration curve is $y = 0.6616x - 0.0088$ and for the other is $y = 0.6412x + 0.22$."

Comment 19 (Figure 3): x is missing.

Changes in the manuscript: x was added.

Comment 20 (line 254). The figure shows two distinctive slopes. Also, how the authors would explain the fact that the 100 ng m-3 standard signal with uncertainty is not within the calibration confidence interval?.

Author's response: Yes, this is what we wanted to show that even if the standards were analysed in the same experiment, there are issues in the lower range as can be seen by the difference in the slopes that is over 30%. The confidence interval was calculated using all 10 calibration points, and being so many in the lower range of the calibration curse, the confidence interval is shorter that in the case in which we would not use the lowest 5 calibration points.

Changes in the manuscript: none

Comment 21 (line 276). What does it mean "extended" uncertainty?

Author's response: It is a mistake. We will change it to "expanded uncertainty".

Changes in the manuscript: "extended uncertainty" was changed to "expanded uncertainty".

Comment 22 (line 312). This uncertainty is quite high. Limit of quantification values should be stated with acceptable uncertainty. Lower values should not be reported since they are not informative.

Author's response: The LOQ is approximately 3 ng m$^{-3}$ with slight variations in different days, analysed by using a carbon trap. Due to the fact that the expanded uncertainty falls above the LOQ, we wanted to include the data, given that there are lots of measurements in this range of concentrations reported in other publications.

Changes in the manuscript: none

**Anonymous referee #2, Short report 2.**

Comments: Fig 1, line 160: it would be useful to give more technical details about the experimental setup. For example, there is no information about the material of the connecting tubes, it only says the "Teflon mixing coil".

Furthermore, the detailed technical parameters of the impinger, which influence the Hg0 separation and the efficiency of the method, are not clear – and should be described in detail. The material of this crucial part is mentioned just by-the-way on line 365.

Also the connection (line 150) from the output T slit to the Zeeman AAS device deserves better description. It would be best for the reader to add detailed photos of the apparatus (to the SI) and to give more technical/construction details, even dimensions, of the main custom made parts.

> Changes in the manuscript: The next section was added: "The impinger is made of quartz and has a height of 46 cm and a diameter of 13.5 cm. All the T-pieces used in the set-up are made of Teflon, along with all the connections until the impinger. Between the impinger and the analyser the connections are made with Tygon tubing and the last T-piece is made of quartz. The quartz frit is placed 2 cm above the bottom of the impinger."